# Perceived Competence, Achievement Goals, and Return-To-Sport Outcomes: A Mediation Analysis

**DOI:** 10.3390/ijerph17092980

**Published:** 2020-04-25

**Authors:** Elyse D’Astous, Leslie Podlog, Ryan Burns, Maria Newton, Bradley Fawver

**Affiliations:** 1College of Health, Department of Health, Kinesiology, and Recreation, University of Utah, Salt Lake City, UT 84101, USA; elyse.dastous@hsc.utah.edu (E.D.); les.podlog@utah.edu (L.P.); ryan.d.burns@utah.edu (R.B.); maria.newton@health.utah.edu (M.N.); 2School of Medicine, Department of Physical Medicine and Rehabilitation, University of Utah, Salt Lake City, UT 84101, USA

**Keywords:** approach-avoidance, collegiate athletes, injury, motivation

## Abstract

The purpose of this study was to explore the potential mediating effect of achievement goals on perceived competence and return-to-sport outcomes among college athletes sustaining a sport injury. Altogether, 75 male and female college athletes from the United States who returned to sport after having missed competition for an average of 3 weeks due to injury, completed valid and reliable inventories measuring perceived competence, achievement goals, and return-to-sport outcomes. Results indicated that task-approach goals significantly mediated the relationship between perceived competence and a renewed sport perspective. These data suggest the importance of promoting competence beliefs and a task-oriented focus among athletes returning to sport following athletic injury. From a practical standpoint, clinicians can foster competence perceptions by integrating progressive physical tests assessing functionality and sport-specific skills/abilities. Furthermore, these data suggest that coaches, physical therapists, and significant others may do well to use language that orients injured athletes towards attaining success as opposed to avoiding failure, to emphasize effort, task completion, and correct form, and to avoid comments that compare athletes to others or to their preinjury standards of performance. From a theoretical standpoint, our mediation findings extend previous achievement goal research into the sport injury domain, further highlighting the importance of task-approach goals.

## 1. Introduction

Over the past 30 years, the profound psychological impact of injury on competitive athletes has been well documented [1,2,3,4]. Of particular relevance to the current study is the growing body of literature exploring the psychosocial aspects of athletes’ return to competition following injury recovery [5]. Athletes transitioning back into sport specific training may experience a lack of competence in their abilities and a variety of apprehensions (e.g., uncertainties about performing to preinjury standards, worries over falling behind competitors, doubts about physical fitness, self-presentational concerns, and re-injury worries) prior to their return to competition [6]. Conversely, returning athletes have also reported positive appraisals about their upcoming return, for example, excitement about obtaining or surpassing pre-injury levels, improving skills, maintaining fitness levels, and preserving athletic identity [6,7,8]. These findings suggest a dichotomy of competence-based appraisals, whereby athletes attempting a return to competition are simultaneously motivated to avoid demonstrating incompetence (i.e., avoid a negative outcome) and to demonstrate competence (i.e., approach a positive outcome; [9]).

Research on the return to competition also reveals that athletes may experience a mixture of adaptive and maladaptive outcomes once they return to competition following injury recovery [1,6]. In an effort to characterize the quality of athletes’ post-injury experiences, Podlog and Eklund [10] developed the return to sport after serious injury questionnaire (RSSIQ). This inventory assesses athlete’s return-to-sport outcomes across two subscales—a “renewed perspective on sport” and “return to sport concerns.” The renewed perspective subscale represents positive perceptions of the return to competition (e.g., greater mental toughness, appreciation for sport), while the return to sport concerns subscale represents negative post-injury outcomes (e.g., heightened competitive anxiety, fears of re-injury, inability to focus during competition). To date, the RSSIQ has been used in several empirical studies on athletic injury [11,12,13] and cited in numerous reviews [14,15,16,17,18]. However, despite advances in knowledge regarding the psychosocial aspects of a return to competition from injury using measures such as the RSSIQ, further theoretically grounded work is needed to uncover the critical psychosocial variables impacting athletes’ return to competition outcomes following injury. Such research is pivotal for informing clinical practice and applied interventions with returning athletes.

### 1.1. Achievement Goal Theory—Return to Sport Following Injury

Given that perceptions of competence may be at the forefront of athletes’ minds as they return to competition after injury [19], and that returning athletes strive to either demonstrate competence or avoid demonstrating incompetence [12], it seems prudent to consider a competence-based theory in examining the relevant psychosocial factors impacting athletes’ return-to-competition outcomes following injury. Elliot’s hierarchical model of approach and avoidance achievement motivation [20], grounded in Achievement Goal Theory (AGT), provides such a model. According to AGT, an achievement goal is conceptualized as the purpose or aim of competence-based action [21] and is posited to regulate how individuals interpret, experience, and act within achievement settings [20]. Competence-based action may be guided by the desire to achieve self-referenced mastery goals focused on personal mastery of skills or by the intent to achieve performance (i.e., ego) goals in which success is defined by performance relative to others or to normative standards. The valence (positive vs. negative) or motivational component of competence refers to the distinction between the motivation to approach positive/desirable events or possibilities, and avoidance motivation, where behavior is directed by an avoidance of a negative/undesirable event or possibility [22]. Thus, in the original 2 × 2 AGT framework, four goals—mastery-approach, mastery-avoidance, performance-approach, and performance-avoidance—are theoretically possible [23,24]. Mastery-approach goals reflect an interest in improving one’s personal performance, while mastery-avoidance goals reflect a desire to avoid performing worse than one aspires to. Performance-approach goals are characterized by a desire to outperform others, while performance avoidance-goals are reflected in the desire to avoid performing worse than one’s peers or normative standards [23,24]. Here, we briefly review relevant empirical evidence prior to introducing Elliot and colleagues’ novel 3 × 2 AGT framework [25] and the current study.

### 1.2. Achievement Goals in Sport—Consequences and Antecedents

Mastery approach (MAp) goals are suggested to be the most adaptive achievement goal and have been shown to positively predict beneficial outcomes such as intrinsic motivation and enjoyment in sport [26], positive affect [27], well-being [28], and performance [29]. Meta-analytic findings have emphasized the importance of MAp thought patterns as a general performance enhancement strategy, indicated by moderate effect sizes [23,24]. Mastery-avoidance (MAv) goals are believed to have a more maladaptive pattern of consequences than MAp goals [30], with evidence suggesting a positive association with amotivation [31], threat appraisals [27], low self-esteem [32], anxiety [33], and negative affect [34].

Performance-approach (PAp) goals reveal an assortment of adaptive and maladaptive outcomes [23,24]. Adaptive outcomes from adopting PAp goals include enhanced perceptions of self-confidence [35], well-being [28], and performance [23], while maladaptive outcomes include threat appraisals [36], self-handicapping [37], emotional and physical exhaustion [32], and negative reactions to imperfection [38]. Performance-avoidance (PAv) goals are conceptualized as the most maladaptive achievement goal, and have been shown to positively predict deleterious outcomes such as anxiety [33], threat appraisals [27], self-handicapping [39], amotivation [40], and negative reactions to imperfection [38].

In addition to the outcomes associated with achievement goals, Elliot’s hierarchical model of approach and avoidance achievement motivation outlines a number of possible antecedents to goal-adoption, including perceived competence (i.e., perceptions of one’s ability or proficiency on particular tasks or endeavors), motive dispositions (e.g., need to achieve and fear of failure), implicit theories of ability, perceptions of motivational climate, and environmental or personal factors (e.g.,) [20]. Elliot contends that these antecedents indirectly modify achievement behavior and outcomes through their influence on achievement goal adoption [41,42]. For instance, the belief that one is competent on a task is expected to increase an individual’s tendency to approach situations in which he or she can improve on their personal performance and/or perform well in relation to others (i.e., mastery or performance approach goals). Approach goals are, in turn, expected to positively influence performance or achievement behaviors.

Researchers in the sport domain have found a number of antecedents for MAp goals including perceived competence [43], mastery climates [44], and incremental beliefs of ability [33]. MAv goals have been negatively predicted by perceived competence [32] and mastery climates [44] as well as positively predicted by fear of failure [45] and performance climates [32]. Antecedents of PAv goals include entity beliefs about sport ability [33], fear of failure [40], and performance climates [43]. Lastly, PAp goals have been predicted by fear of failure [40], performance climates [44], and perceived competence [40]. Taking into consideration that perceptions of competence are especially relevant to athletes returning to competition after injury, it seemed prudent to examine perceived competence as an influential antecedent to achievement goal adoption among this population.

### 1.3. The 3 × 2 Conceptual Model of Achievement Goals

Recently, the 2 × 2 achievement goal model was expanded by Elliot [25] into the 3 × 2 model by crossing the object of competence (task—self—other) with the valence of competence (positive, approach—negative, avoidance; [25,46]). Task-referenced goals focus on efficacy in accomplishing the set task, while self-referenced goals focus on current performance relative to past or potential future performance. Other-referenced goals are synonymous with performance goals from the 2 × 2 model, where the focus is on how one does relative to specific others or normative standards. When the dimensions are crossed in this new conceptualization, a total of six resultant goals are possible: (1) task-approach (TAp; to demonstrate task-referenced competence); (2) self-approach (SAp; to demonstrate self-referenced competence); (3) other-approach (OAp; to demonstrate other-referenced competence); (4) task-avoidance (TAv; to avoid demonstrating task-referenced incompetence); (5) self-avoidance (SAv; to avoid demonstrating self-referenced incompetence); and (6) other-avoidance (OAv; to avoid demonstrating other-referenced incompetence; [46]). Differentiation of task- and self-based standards is clearly relevant in the sport domain given that in any sport activity or physical task, individuals may focus on whether they are (or are not) accomplishing the task, on how they are doing relative to how they have done in the past or how they may do in the future, and/or lastly on how they are doing relative to others [46].

A plethora of literature exists on the 2 × 2 dichotomous model of achievement goal theory in sport [12,24,42,46], although none in the context of sports injury. Moreover, aside from a handful of studies in educational [47,48,49] and sport settings [50,51], there is a dearth of research conducted using the 3 × 2 model of achievement and no work currently focused on sport injury. The 3 × 2 model has particular relevance for recovery from sport injury and return to play, given that injured athletes might be differentially motivated to master goals relevant to the task (i.e., performing the relevant rehabilitation tasks properly or sport-specific skills) and relevant to themselves (i.e., achieving pre-injury physical ability and skill level). Moreover, a number of qualitative studies have described at least one of the six achievement goals in the 3 × 2 model in relation to return to sport from injury. For instance, Driediger, Hall, and Callow [52] reported an injured wrestler’s example of an other-approach goal (i.e., “wanted to get back and beat [his competition]”, p. 265), while another athlete described a task-avoidance motivation when returning, saying, “I imagine myself at practice and completely sucking… or being horrible, or out of shape and I don’t want that to happen” ([52], p. 267). In Podlog and Eklund’s [53] qualitative study of high level athletes’ perceptions of success when returning to sport following injury, they noted that some athletes were motivated to contribute to the team’s success and to receive positive feedback from coaches (i.e., task approach). Therefore, the 3 × 2 model might provide superior sensitivity with respect to approach-avoidance motivation during return to sport following injury, and could help explain the link between perceptions of competence and return-to-sport outcomes in injured athlete populations.

### 1.4. The Current Study

Four main conclusions can be drawn from the aforementioned research: (a) athletes on the cusp of a return to competition after injury may have diminished perceptions of competence [6,19]; (b) athletes may experience competing approach and avoidance behavioral tendencies prior to their return to competition [6,8,53]; (c) athletes may experience a range of positive and negative return-to-competition outcomes [1,6,10]; and (d) achievement goals can lead to positive and negative outcomes in sport [23,24] and that the adoption of achievement goals is predicted by a number of possible antecedents, including perceived competence [40]. The purpose of this study was to use Elliot’s hierarchical model of approach and avoidance achievement motivation [25,46] to explore relationships between perceived competence, achievement goals, and return-to-competition outcomes among collegiate athletes who had returned to competition following a serious sport injury.

Within our overall purpose, we sought to examine three specific aims. First, we tested whether athletes’ perceived competence predicted return concerns and a renewed sport perspective. We first hypothesized that perceived competence would positively predict a renewed sport perspective and negatively predict return-to-sport concerns. Second, we aimed to test whether achievement goals predicted return-to-sport outcomes. Considering the minimal research on the 3 × 2 model of achievement goal theory, we had little support for an a priori hypothesis regarding return-to-sport outcomes as a function of competence definition (i.e., task-, self-, and other-referenced), so we did not forward a directional prediction. The final specific aim of this study was to examine whether achievement goals partially mediated the relationship between perceived competence and return-to-sport outcomes. Assuming that both perceived competence (antecedent) and achievement goals (mediators) were significant predictors of return-to-sport outcomes, our third hypothesis was that achievement goals would partially mediate the relationship between perceived competence and return-to-sport outcomes. As achievement goals are defined by the aim and direction of one’s competence-based pursuits, as part of our third hypothesis, we also expected perceived competence to have a direct influence on achievement goal adoption.

## 2. Materials and Methods

### 2.1. Participants

Seventy-five male and female college athletes from the United States (males = 45; *M*age = 21; *SD* = 2.15) competing in team (basketball: *n* = 6, football: *n* = 7, hockey: *n* = 1, lacrosse: *n* = 11, rugby: *n* = 2, soccer: *n* = 14, softball: *n* = 1, volleyball: *n* = 5, ultimate frisbee: *n* = 8, cheer: *n* = 1, field hockey: *n* = 1, and baseball: *n* = 1) and individual sports (swimming: *n* = 4, powerlifting: *n* = 1, skiing: *n* = 1, track and field: *n* = 6, dance: *n* = 1, cross country: *n* = 2, wrestling: *n* = 1, gymnastics: *n* = 1, and golf: *n* = 1) volunteered to participate in the study. To be eligible for study involvement, participants were required to meet several criteria. First, prospective participants needed to compete on a university team or club at the National Collegiate Athletic Association (NCAA) Division I-III level, or be a member of a National Association of Intercollegiate Athletics (NAIA) team—the latter league representing a collection of small colleges and universities in North America. Athletes competing at an NCAA Division I level, represent the highest level of collegiate competition in the United States. Similarly, those competing at the NCAA Divisions II-III and at the NAIA levels also participate at a very competitive–albeit slightly lower level–than their Division I counterparts. Second, athletes must have sustained a self-reported injury (incurred in or outside of sport) which required a minimum 3-week absence from sport specific training and competition, a time-loss synonymous with a severe injury [6,10]. Third and finally, the individual must have returned to competition within the past two years. The latter criterion was designed to mitigate memory loss or recall bias associated with injuries occurring in the distant past. Any individual not meeting these three eligibility criteria were excluded from study participation. Our final sample included participants competing at club (*n* = 21), NCAA Division I (*n* = 17), NCAA Division II (*n* = 18), NCAA Division III (*n* = 3), NAIA (*n* = 10), and Junior College (an institution offering courses for two years beyond high school, either as a complete training or in preparation for completion at a four-year degree granting college; *n* = 3), with 3 unreported levels of competition. Participants represented a convenient sample returning from an injury that on average prevented them from participating in regular sport training and/or competition for approximately 20 weeks (*M* = 19.8 weeks, *SD* = 23.0, range = 3 to 133 weeks). Athletes experienced a variety of injury types including (but not limited to): ACL, AC joint, hamstring, foot, wrist, and concussions.

### 2.2. Measures

#### 2.2.1. Perceived Competence

A modified version of the perceived competence subscale from the Intrinsic Motivation Inventory (IMI; [54]), a valid and reliable instrument [55], was used to measure participants’ perceptions of competence after receiving clearance to return to competition. Following the stem “After receiving clearance to return to regular training and/or competition,” sample items included, “I felt I wouldn’t be able to compete in my sport very well” and “I thought I would be pretty good at my sport.” Participants indicated their level of agreement with each of the 5 items each on a 5-point scale ranging from strongly disagree (1) to strongly agree (5).

#### 2.2.2. Achievement Goals

The 3 × 2 Achievement Goal Questionnaire for Sport (AGS-S; [46]) was used to measure the degree to which participants endorsed different achievement goals following their most recent serious injury prior to making their return to competition. The stem “In sport, my goal is…” from the original scale was changed to “Upon returning to competition after injury my goal was…” The 3 × 2 AGQ-S measures six goals, each with three items: task-approach (TAp; e.g., “to perform well”), self-approach (SAp; e.g., “to do better than I usually do”), other-approach (OAp; e.g., “to do better than others”), task-avoidance (TAv; e.g., “to avoid performing badly”), self-avoidance (SAv; e.g., “to avoid having worse results than I had previously”), and other-avoidance (OAv; e.g., “to avoid doing worse than others”). Participants responded on a 7-point scale ranging from strongly disagree (1) to strongly agree (7). The 3 × 2 AGQ-S has demonstrated good internal consistency (α > 0.7) and construct validity [46].

#### 2.2.3. Return to Sport Outcomes

The Return to Sport After Serious Injury Questionnaire (RSSIQ; [10]) was used to assess athlete perceptions of their post-injury performances, that is, the time-frame following their return to competition after injury. The stem, “Within my first season returning to sport after injury…” prefaced each of the 15 items from the RSSIQ. Ten items of the RSSIQ represent return concerns (e.g., “My confidence in performing challenging skills and techniques has been lower;” “My fear of reinjury has interfered with performances;” “My anxiety about competing has been greater;” “My ability to perform has been affected by my injury”) and 5 items represent a renewed perspective on sport (e.g., “My enjoyment of practice and competition has been greater;” “My motivation for sport success has been greater;” and “My understanding about how to train/compete has been better”). Participants indicated their level of agreement to each item on a 7-point scale ranging from strongly disagree (1) to strongly agree (7). The RSSIQ has demonstrated adequate internal consistency as well as initial construct validity [10].

### 2.3. Procedures

After receiving Institutional Review Board (IRB; i.e., ethics) approval (#00086873), coaches and/or captains, athletic trainers, and sport science professors were contacted and invited to share a description of the nature and aims of the study with potential participants. In an effort to increase the generalizability of the findings, we recruited participants from a variety of institutions. Eligible participants were then referred to the primary investigator who provided them the survey and an informed consent cover letter, either via an online link or a paper copy. Consent was considered obtained upon completing the survey online or returning the paper survey to the primary investigator. Completion of the survey took approximately 10 min.

### 2.4. Data Processing and Analyses

Item averages for the IMI and each RSSIQ and 3 × 2 AGQ-S subscales were calculated, and data were cleaned and screened prior to conducting the main analyses. A total of six missing data points from 3 different participants were identified and subsequently replaced with the series mean. The initial data analyses involved calculating descriptive statistics, internal consistency scores, and bivariate Pearson product-moment correlations across all study variables. Bivariate correlations were determined to be weak (*r* <0.30), moderate (*r* = 0.30–0.60), or strong (*r* > 0.60) based on established criteria [56].

Two bootstrap analyses were subsequently conducted to explore the potential mediating effect of task-approach goals in explaining the link between perceived competence and return to sport outcomes. Bootstrapping is a robust analytic technique that can be applied to non-normal data, thus making the identification of outliers inessential [57]. The bootstrap mediation analyses consisted of employing the Preacher and Hayes method [57] to calculate direct effects, indirect effects, and total effects across 1000 iterations.

The indirect effect (IE) was the average mediated effect between perceived competence and return-to-sport outcomes via a task-approach goal mediator; the direct effect (DE) was the average effect of perceived competence on the return to sport outcome without use of an achievement goal mediator, and the total effect (TE) was the entire effect that perceived competence had on the return to sport outcomes (TE = DE + IE). Other parameters of interest included the total variance in the return to sport outcome explained by a respective mediation model (including the primary antecedent and mediator), which was calculated using the coefficient of determination (*R*^2^), in addition to the proportion of the total effect attributed through a respective partial mediation mechanism (*P*_M_ = indirect effect/total effect).

A partial mediation between perceived competence and renewed perspective was expected, with moderate-to-large effect sizes for the direct effect (partial *r* = 0.3–0.6). Based on established criteria to detect partial mediation using a bootstrapped analysis [58], we expected 59-78 participants would be adequate to achieve 80% statistical power. A post-hoc assessment of observed effect sizes in the current study using MedPower [59], indicated that the current sample achieved 75% power. All analyses had an initial alpha level of *p* < 0.05 and were carried out using the SPSS v.23.0 statistical software package (Armonk, NY, USA) and PROCESS macro v.2.16 [60].

## 3. Results

### 3.1. Descriptive Statistics and Bivariate Correlations

Descriptive statistics, correlations, and internal reliability scores are presented in Table 1. Participants exhibited high scores on all six achievement goals (*M* > 5; range = 1–7) as well as relatively high scores for perceived competence (*M* = 3.38, range = 1–5). Mean scores on the RSSIQ indicate that the participants experienced higher levels of a renewed perspective (*M* = 5.20, *SD* = 1.05) than they did return concerns (*M* = 4.13, *SD* = 1.49) with a difference that was statistically significant (*t*(76) = 4.62, *p* < 0.001). Internal reliability scores among study variables were deemed acceptable (α range = 0.71–0.92) after removing one item from the self-avoidance goals subscale.

As indicated in Table 1, perceived competence was moderately positively correlated with a renewed perspective (*r* = 0.36, *p* < 0.01), while a moderate negative correlation was observed between perceived competence and return concerns (*r* = −0.54, *p* < 0.01). Perceived competence exhibited a moderately positive correlation with task-approach (TAp) goals (*r* = 0.30; *p* < 0.01), whereas null findings were observed for the relationships between perceived competence and the other five goals. TAp goals were moderately and positively correlated to a renewed perspective (*r* = 0.44; *p* < 0.01) and weakly and negatively correlated to return concerns (*r* = −0.22; *p* < 0.05). Other-approach (OAp) goals were weakly and positively correlated to a renewed perspective (*r* = 0.28; *p* < 0.05) and weakly and negatively correlated to return concerns (*r* = −0.24; *p* < 0.05). No other significant correlations between achievement goals and return-to-sport outcomes were observed. A renewed perspective was moderately and negatively correlated to return concerns (*r* = −0.39; *p* < 0.01). Weak to strong correlations emerged among the six achievement goals (*r* = 0.15 to *r* = 0.74).

### 3.2. Mediation Analysis

Based on the correlation matrix, task-approach goals were the only goals significantly correlated to the conceptualized antecedent (i.e., perceived competence) and two return-to-sport outcomes (i.e., renewed perspective and return concerns). Thus, only two mediation analyses were conducted with *perceived competence* entered as the independent variable and *task-approach goals* as the mediator variable for both analyses. The dependent variable for the first analysis was a *renewed perspective*, and the dependent variable for the second analysis was *return concerns*.

As depicted in Figure 1, the results of the first mediation analysis showed that there was a statistically significant indirect effect of perceived competence on a renewed perspective through the adoption of task-approach goals (*F*(1, 73) = 9.33, *p* = 0.003). The standardized parameter estimates of this model included a statistically significant indirect effect (i.e., a-path*b-path) (IE = 0.1483, BCa CI [0.048, 0.305], *p* = 0.016), a statistically significant direct effect (i.e., c’-path, Figure 1) (DE = 0.323, BCa CI [0.019, 0.628], *p* = 0.038), and a statistically significant total effect (i.e, c-path, Figure 1) (TE = 0.472, BCa CI [0.164, 0.779], *p* = 0.031). Statistically significant pathways were also documented for perceived competence predicting task-approach goals (i.e., a-path, Figure 1**)** (*p* = 0.004), and task-approach goals predicting a renewed perspective (i.e., b-path, Figure 1) (*p* = 0.001). The mediation pathway accounted for 31.4% of the TE (*P*_M_ = 0.314) with the full mediation model (using both antecedents) explaining 25.4% of the total variance in renewed perspective (*R*^2^ = 0.254). The second mediation model with perceived competence, task-approach goals, and return concerns as the outcome was not statistically significant.

## 4. Discussion

Grounded in the hierarchical model of achievement motivation [20], the present study examined: (a) whether formerly injured collegiate athletes’ perceived competence in their sporting abilities predicted return-to-sport outcomes; (b) whether achievement goals predicted return-to-sport outcomes; and finally (c) if both the antecedent (i.e., perceived competence) and mediator (i.e., achievement goals) demonstrated a significant relationship with return-to-sport outcomes, whether achievement goals partially mediated the relationship between perceived competence and return-to-sport outcomes. Findings revealed moderate support for these hypotheses, with task-approach goals partially mediating the relationship between perceived competence and a renewed sport perspective. Results also suggest that athletes who believe themselves to be capable and proficient in their sport are driven by goals to perform well, to be effective, and to obtain good results. These goals, in turn, facilitate the perception of beneficial outcomes such as greater enjoyment, mental toughness, and motivation for sports success. Since the significance of a mediation analysis is determined by the indirect effect (i.e., a-path*b-path), the significant paths from antecedents to goals (i.e., “a-path”) as well as from goals to outcomes (i.e., “b-path”) are discussed along with other significant findings.

Our first hypothesis that perceived competence would positively predict a renewed perspective and negatively predict return concerns was supported. These findings make intuitive sense given that high perceptions of sport ability are positively associated with greater enjoyment, mental toughness, and sport proficiency and inversely related to low confidence, anxiety, and poor performance [23,26,27,29,61,62]. In their study of high-level adult athletes who returned to sport from an injury, Podlog, Lochbaum, and Stevens [12] found that competence positively predicted a renewed perspective, but failed to negatively predict return concerns. Athletes in the current study reporting higher levels of a renewed perspective compared to return concerns after injury is also consistent with previous research [10]. This is a promising finding, because although injury is an unpleasant event, many athletes are able to recognize this setback as an opportunity for psychological growth [63]. Additionally, the negative correlation between return concerns and a renewed perspective suggests that the two factors are accounting for different (i.e., positive and negative) psychological outcomes associated with a return to sport after injury and provides additional support for the RSSIQ [10].

The bifurcation of mastery-approach goals from the 2 × 2 to the 3 × 2 achievement goal model resulted in task-approach, but not self-approach, goals significantly predicting return-to-sport outcomes. This finding provides support for the 3 × 2 achievement goal model and is in accordance with Mascret and colleagues’ [46] position that an exhaustive cross of both the definition and motivational orientation of competence allows for greater precision and rigor in explaining the nature of achievement motivation in the sport domain. The differentiation of task- and self-based standards is conceptually relevant to the domain of sport injury because individuals may focus on whether they are (or are not) accomplishing the task (TAp/TAv) or alternatively, they may focus on how they are doing relative to how they have done in the past (preinjury) or how they may do in the future (SAp/SAv). The latter two foci may lead to frustration and/or amotivation if athletes are failing to live up to these standards during rehabilitation. Although this study provided partial support of the relevance of the 3 × 2 AGT model [25] in the context of sport injury, additional research is needed to gain a more extensive understanding of the unique antecedents and outcomes of each achievement goal adoption.

We offered no directional hypothesis concerning the relationship between achievement goals and return-to-sport outcomes (the “b-path”), although we suggested that avoidance goals may be adaptive by encouraging athletes to take precautions in order to avoid aggravating their injury. Although individuals in sports contexts tend to adopt mastery-approach goals more frequently than avoidance goals, such a conceptualization of avoidance as being potentially adaptive is supported by recent meta-analytic findings [24] and has some merit, particularly considering the context of overall outcomes when contrasting approach vs. avoidance achievement. However, in the current study, no significant relationships were found between avoidance goals and return-to-sport outcomes. On the other hand, both task- and other-approach goals were positively associated with a renewed perspective and negatively associated with return concerns. The only significant mediation analysis was the “b-path” from task-approach goals to a renewed perspective. This result is consistent with previous research that found approach goals to positively predict adaptive outcomes such as enjoyment, motivation, and performance [29,31,37,64,65,66]. Task-approach goals thus appear to be equally facilitative for athletes returning from sports injury as for athletes who have not sustained an injury. Coaches, athletic trainers, teammates, and significant others could encourage the adoption of task-approach goals among returning athletes by emphasizing effort, task-completion, correct form, and consistency. Conversely, comparisons to others (i.e., OAp/OAv goals), comparison to one’s preinjury self, or comparisons to one’s perceived potential had the athlete not sustained an injury (i.e., SAp/SAv goals) should be de-emphasized, as such goals could be detrimental to athletes’ motivation and enjoyment should they fail to “match up” when returning to practice and competition.

It is interesting to explore potential explanations for why none of the avoidance goals predicted either return-to-sport outcome. In their review, Van Yperen, Blaga, and Postmes [67] suggested that given the inherent social comparison and competitiveness of sport, individuals adopting performance-avoidance goals in the sport domain—as opposed to the work or education domains—may not experience a lack of focus, effort, or persistence. Similarly, among athletes returning to sport after injury, avoidance goals may not denote a negative connotation because not being worse than you were before (SAv), not being worse than others (OAv), or not being ineffective (TAv) may be regarded positively, especially given the setbacks of injury. Thus, avoidance goals can potentially be interpreted as neutral, as opposed to detrimental, in terms of predicting return-to-sport outcomes among injured athletes. Elliot and Conroy [21] warn, however, that avoiding a negative outcome is an inherently maladaptive form of regulation. Even when seemingly benign or facilitative in the short-term, avoidance goals (e.g., fear of failure, fear of re-injury) are likely associated with long-term adverse effects such as negative affect, undermined intrinsic motivation, and deteriorated performance [21]. Lastly, Lochbaum and colleagues underscored the negligible effects of avoidance goals on antecedents and outcomes in their recent meta-analytic review [24]

We found some support for our third hypothesis with task-approach goals (i.e., the mediator) partially mediating the relationship between perceived competence (i.e., the antecedent) and a renewed sport perspective (i.e., the outcome). Moreover, perceived competence was also positively associated with task-approach goals, indicating a significant “a-path,” which is consistent with previous research demonstrating that high perceptions of competence positively predict approach goals [40,43,62]. This makes intuitive and conceptual sense given that when individuals believe they are proficient and capable, they will likely be motivated to demonstrate that competence in achievement settings and be directed by the positive outcomes of their efforts [22]. Given that the mediation model accounted for just over 25% of the total variance in a renewed perspective on sport, other psychological factors appear to play a role in athletes’ return to sport outcomes. For example, previous research suggests that confidence in one’s return to sport capabilities, re-injury anxiety, social support, and autonomy-supportive environments may all account for the quality of athletes’ post-injury performances and return to sport experiences [5,6,7,12].

Results of the current study provide support for the 3 × 2 Hierarchical Model of Approach and Avoidance Achievement [25], and are similar to those of Mascret and colleagues [46] who found that perceived competence was positively related to task-approach- and other-approach goals but unrelated to self-approach goals. An interpretation of the null results between perceived competence and SAp goals may be that returning athletes with high and low perceptions of competence alike strive for improvement (SAp; [46]) or that self-presentation (i.e., ego) concerns encourage athletes to avoid failure regardless of perceptions of actual competence. These findings suggest the value in promoting competence perceptions (e.g., review progress, visualize effective performance, analyze videos of positive performance) among athletes returning to sport following injury.

### 4.1. Practical Implications

Overall, results from the current study highlight the importance of taking steps during rehabilitation to ensure athletes perceive themselves to be capable of meeting the physical and psychological demands of their sport prior to making their return to practice and/or competition. Such steps may include progressive physical tests assessing functionality and sport-specific skills/abilities as well as psychological skill interventions such as goal-setting, imagery, self-talk, attentional focus, and emotion regulation [68]. For instance, goal-setting and imagery strategies which have demonstrated efficacy in sport injury rehabilitation [69,70] should focus on successful mastery and performance of rehabilitation and sport-specific skills and engagement in exercises designed to mitigate future injuries. In so doing, practitioners can help athletes feel that they are equipped to deal with the myriad challenges inherent in a return to sport following injury.

That athletes reported higher levels of a renewed sport perspective compared to return concerns after injury is also consistent with previous research [10]. As such, practitioners may wish to point out that injury may provide opportunities for personal growth and enhanced post-injury performance. Consistent with past research [71], clinicians can highlight the fact that athletes who experience positive post-injury benefits adopt certain behavioral strategies, for example, seeking required social support, developing stronger relationships with relevant others, or using the time away from sport to nurture other interests. In an effort to promote a renewed sport perspective, practitioners can also encourage injured athletes to use the rehabilitation period as an opportunity to improve sport strategy, technical aspects of one’s sport, and previously neglected areas of one’s sport performance.

Findings regarding the benefits of task approach goals indicate that coaches, sport medicine practitioners, and health care providers can help injured athletes by adopting language that: (a) orients athletes towards attaining success as opposed to avoiding failure; (b) emphasizes effort, task completion, and correct form; (c) avoids comparing athletes to others or to their preinjury standards of performance. These suggestions are bolstered by Lochbaum and colleagues 2015 meta-analytic findings highlighting the deleterious implications of avoidance goals on performance [23], although Lochbaum et al.’s recent meta-analysis calls into question the conceptual clarity offered by avoidance goal constructs compared to approach goals when considering performance outcomes [24].

### 4.2. Limitations and Future Directions

The present study was the first to explore the utility of the 3 × 2 hierarchical model of achievement motivation in the context of sport injury. Although these findings have practical implications and contribute to the literature on achievement motivation, sport, and sport injury, there are several limitations worth acknowledging. First, the cross-sectional design of the study limits the generalizability of the findings as well as the ability to determine causality. The nature of sport injury research makes it difficult to recruit large homogenous samples, but future research would benefit by controlling for injury type and time loss. Along these lines, it should be noted that differences in injury type, location, or severity may have variable implications for athletes’ competence perceptions, their approach-avoidance tendencies and subsequent return to sport outcomes. For example, a soccer player who experiences a lower body injury such as an ACL tear, versus an upper body injury (e.g., wrist injury), may have greater avoidant tendencies (e.g., wanting to avoid re-injury), given the importance of lower body movements (e.g., pivoting, twisting, cutting) to overall soccer performance. As such, researchers are encouraged to examine the potential moderating impact of injury type, location or severity on perceptions of competence, achievement goals, and return to sport outcomes. Second, recruitment of a larger sample size may enable researchers to more fully explore the potential impact of sport-specific (team versus individual sport, combat versus non-combat sports, norms of the sporting sub-culture) or demographic (e.g., gender or age) factors on psychological variables of interest and return to sport outcomes. For example, an injured quarterback in American football may experience different pressures (e.g., team pressures to return to sport, scholarship concerns, intense media scrutiny) than an athlete competing in an individual, less high-profile sport (e.g., powerlifting, cross-country), all of which may exert differential impacts on the injured athlete’s psychological status (e.g., competence perceptions, fear of re-injury, goal-orientations) and return to sport outcomes. Researchers are encouraged to explore such possibilities through the recruitment of robust samples of athletes in different sports, or alternatively, examine the psychological implications of injury among athletes in one particular sport at a specific competitive level. Third, the retrospective design could be criticized for increasing the likelihood for memory decay and recall bias, although we attempted to minimize this limitation by introducing a 2-year criterion within which athletes must have returned to their sport after injury. One strength of using a retrospective design for this study, however, was that it permitted athletes to reflect on their overall injury experience and to ascertain the outcomes of their injury. In future, researchers should aim to implement longitudinal designs, which would minimize memory biases but still allow athletes to provide insights into their perceived outcomes. Longitudinal designs would also enable exploration of whether pre-injury factors (e.g., personality traits) influenced athletes post-injury adoption of certain achievement goals and/or their approach/avoidance tendencies as they neared a return to sport. Additionally, such an approach would also provide the opportunity to follow injured athletes who never return to sport and explore whether their perceptions of competence and achievement goals contribute to non-return to sport. Fourth, research can remove potential shared method variance in independent and dependent variables due to self-reporting by using additional measures of objective performance and informant reports from coaches, athletic trainers, teammates, and significant others to assess return-to-sport outcomes. Fifth, although multiple comparisons were made within the correlation matrix, alpha level was not adjusted in order to minimize potential Type II error rates. However, the approach of not adjusting the alpha level may increase the risk of committing a Type I error, thus caution should be made when making inferences from the correlation matrix [72].

## 5. Conclusions

The findings from this study indicate a promising avenue for future research by using Elliot’s 3 × 2 hierarchical model of approach and avoidance achievement motivation to reveal connections between existing research on psychosocial factors associated with sport injury and return-to-sport outcomes. To reiterate the practical implications from these data, coaches, physical therapists, and significant others may do well to use language that orients injured athletes towards attaining success as opposed to avoiding failure, to emphasize effort, task completion, and correct form, and to avoid comments that compare athletes to others or to their preinjury standards of performance. Other empirically supported techniques such as use of goal-setting, imagery, and positive self-talk may also promote task-oriented approach goals and positive return to sport outcomes [69,70]. Continued empirical efforts in this area may directly inform interventions targeted at modifying achievement goals and their antecedents, in an effort to improve injured athletes’ return-to-sport experiences.

## Figures and Tables

**Figure 1 ijerph-17-02980-f001:**
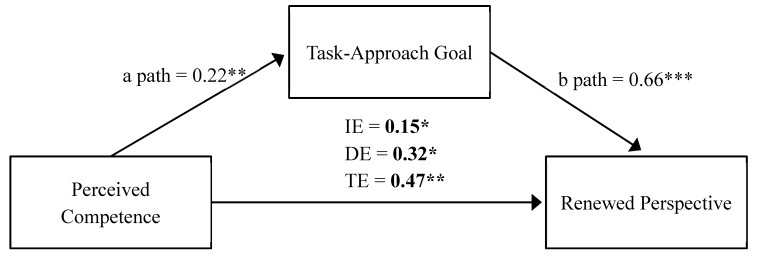
Mediation analysis predicting a renewed perspective. IE = Indirect Effect; DE = Direct Effect; TE = Total Effect; bold indicates statistical significance: * *p* < 0.05, ** *p* < 0.01, *** *p* < 0.001.

**Table 1 ijerph-17-02980-t001:** Descriptive statistics (mean, standard deviation, range), internal consistencies, and correlations among measures of perceived competence, achievement goals, and return-to-sport outcomes.

Variable	*M*	*SD*	Observed Range	Cronbach’s *α*	*r*							
					1	2	3	4	5	6	7	8
1. Perceived competence	3.38	0.77	1.80–5.00	0.78	---							
2. Task-approach goals	6.22	0.55	5.00–7.00	0.76	0.30 **	---						
3. Task-avoidance goals	5.71	1.37	1.33–7.00	0.92	0.14	0.31 **	---					
4. Self-approach goals	5.28	1.18	2.00–7.00	0.77	0.14	0.45 **	0.19	---				
5. Self-avoidance goals	5.42	1.27	1.00–7.00	0.75	0.02	0.21	0.67 **	0.36 **	---			
6. Other-approach goals	5.31	1.27	1.00–7.00	0.86	0.13	0.39 **	0.24 *	0.45 **	0.43 **	---		
7. Other-avoidance goals	5.08	1.43	1.00–7.00	0.84	0.04	0.15	0.72 **	0.18	0.74 **	0.44 **	---	
8. Return concerns	4.13	1.52	1.00–6.50	0.92	−0.54 **	−0.22 *	0.11	−0.08	−0.09	−0.24 *	0.04	---
9. Renewed perspective	5.20	1.01	2.40–7.00	0.71	0.36 **	0.44 **	0.08	0.18	0.05	0.28 *	−0.02	−0.39 **

*Note*. Bold indicates statistical significance: * *p* < 0.05, ** *p* < 0.01.

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
