# Peer review of "Perceived Competence, Achievement Goals, and Return-To-Sport Outcomes: A Mediation Analysis"

_ijerph, 2020, doi:10.3390/ijerph17092980_

Round 1

Reviewer 1 Report

Thankyou for this paper, it is an interesting read and will be of benefit to coaches, medical staff wanting to improve post injury outcomes of their players.

I have a couple of notes I would like to add however:

Line 68 – 69, “and that returning athletes strive to either demonstrate competence or avoid demonstrating incompetence” is there any method to differentiate between the two pre-injury, such as certain personality traits?

Limitations to consider in more depth:

  1. Each sport is different and injury may affect one sport more than another due to competition replacing a player, coach pressure etc. Differing psychological pressure on certain sports maybe higher due to financial success and limelight on certain sports. Further understanding and highlighting the benefit of recruiting a group for one sport at one particular level/league whilst not applicable to ALL sport would be far more powerful to that particular sport, this could be carried through to a soccer, basketball etc. only study.
  2. Differences in injury affecting the sport, a wrist injury in soccer whilst taking the player out for 3w as stated in the study may be easier to return to play for the player compared with an ACL injury that whilst the player returns to the sport may have longer lasting physical and psychological effects. As knee injury would be far more common than wrist injuries in soccer perceived fear of reinjury would be applicable.
  3. Having a higher participant number would allow for a superior distinction of differences between sex and age affecting results if a larger all sports/injury study was considered.

Author Response

1.Each sport is different and injury may affect one sport more than another due to competition replacing a player, coach pressure etc. Differing psychological pressure on certain sports maybe higher due to financial success and limelight on certain sports. Further understanding and highlighting the benefit of recruiting a group for one sport at one particular level/league whilst not applicable to ALL sport would be far more powerful to that particular sport, this could be carried through to a soccer, basketball etc. only study.

Consistent with the reviewer’s point, we have added in a point regarding the value of a larger sample size in exploring the impact of sport specific and demographic factors on constructs of interest on lines 494-505.

2. Differences in injury affecting the sport, a wrist injury in soccer whilst taking the player out for 3w as stated in the study may be easier to return to play for the player compared with an ACL injury that whilst the player returns to the sport may have longer lasting physical and psychological effects. As knee injury would be far more common than wrist injuries in soccer perceived fear of reinjury would be applicable.

The reviewer raises an important point regarding the impact of injury type/location/severity on psychological constructs of interest. We have added this point to our discussion of the study limitations and suggestions for future research on lines 486-494: “Along these lines, it should be noted that differences in injury type, location, or severity may have variable implications for athletes’ competence perceptions, their approach-avoidance tendencies and subsequent return to sport outcomes. For example, a soccer player who experiences a lower body injury, such as an ACL or MCL tear versus an upper body injury (e.g., wrist injury) may have greater avoidant tendencies (e.g., wanting to avoid re-injury), given the importance of lower body movements (e.g., pivoting, twisting, cutting) to overall soccer performance. As such, researchers are encouraged to examine the potential moderating impact of injury type, location or severity on perceptions of competence, achievement goals, and return to sport outcomes.”

3. Having a higher participant number would allow for a superior distinction of differences between sex and age affecting results if a larger all sports/injury study was considered.

As indicated, we have added in a point regarding the value of a larger sample size in exploring the impact of sport specific and demographic factors on constructs of interest on lines 494-505.

Reviewer 2 Report

This paper is exceptionally well written and I enjoyed the narrative. It addresses an important, emerging area of research and it is encouraging to see the authors employ a methodologically pragmatic approach when embarking on an exploratory study. I have very few general concerns with the paper – many of the comments I originally had were addressed frankly in the limitations section – but there are a few specific areas where I think the paper could be improved, as follows:

  1. The introduction is far too long and delves deeper into the underpinning theory than is strictly necessary to build a rationale for the study. I recommend paring it down to the essential information a reader would need to be able to establish the appropriateness of the study’s aims and hypotheses. The rest becomes a shade didactic and would be better placed in the discussion to elaborate on the potential mechanisms behind the findings.
  2. Lines 49 – 50:  This sentence reads, “athletes may experience a variety of positive and negative experiences.” This could be rephrased for better flow.
  3. Please clarify the recruitment strategy – were these athletes identified from a single university program (my assumption is they were), and if so, were they treated through the same medical clinic? There could be systematic environmental factors in this setting that would contribute to potential selection biases (eg, self-referral) or measurement biases (eg, supportive therapists promoting high achievement goal or competence scores), leading to over- or under-estimation of the magnitude of the observed relationships. A comment about this in the methods would help the reader assess the likelihood of such issues and give further insight into the generalisability of the results.
  4. Lines 315-316: are means of 5.2 and 4.1 really different on a 7-point scale? Given that participants respond using whole numbers and no estimates of variability are presented around these means, I’m not convinced that participants experienced meaningfully higher levels of renewed perspective vs concern. I suggest including a measure of variability here – SD or confidence intervals – and specifying whether this is a statistically vs practically meaningful difference.
  5. Line 321: “positively” should be “positive”
  6. The mediation model accounts for roughly one quarter of the total variance in renewed perspective. In the discussion it would be useful to consider what else might be involved in that relationship and the remaining 75%.
  7. The conclusion makes quite a point about the benefit of using language focused on attaining success vs avoiding failure. This is a valid suggestion but without understanding how social support and intrinsic vs extrinsic motivations interact when establishing perceived competence and/or goal orientations, this is a very specific bit of advice. Language choice is, as the authors have pointed out, one suggestion amongst many empirically-supported intervention techniques and it would be prudent to include the evidence informed ones (goal setting, imagery, etc) in the take home message as well.

Author Response

This paper is exceptionally well written and I enjoyed the narrative. It addresses an important, emerging area of research and it is encouraging to see the authors employ a methodologically pragmatic approach when embarking on an exploratory study. I have very few general concerns with the paper – many of the comments I originally had were addressed frankly in the limitations section – but there are a few specific areas where I think the paper could be improved, as follows.

Thank you for the positive comments on the paper. We have addressed each of the specific comments in the line by line responses and in the manuscript. Changes to the manuscript appear in red.

1.The introduction is far too long and delves deeper into the underpinning theory than is strictly necessary to build a rationale for the study. I recommend paring it down to the essential information a reader would need to be able to establish the appropriateness of the study’s aims and hypotheses. The rest becomes a shade didactic and would be better placed in the discussion to elaborate on the potential mechanisms behind the findings.

Consistent with the reviewer’s suggestion, we have trimmed the introduction, specifically the sections cited below.  “In the present study, we address these concerns by examining how competence-based appraisals influence return-to-sport outcomes, and whether that relationship is mediated by achievement goals. (lines 64-66).”

“In sum, both performance–approach and –avoidance goals, demonstrate considerable variability with respect to performance and well-being outcomes in the literature [25, 26], lines 106-108.”

“Within the hierarchical model, the terms antecedents and consequences are not meant to imply causality, but rather are used to communicate the proposed nature of the relationships between variables [27]. Lines 119-121”

“Such an absence is likely attributable to the relatively recent construction of the 3 x 2 achievement goal questionnaire (3 x 2 AGQ; [27]) and the even more recent adaptation of the measure for sports contexts (3 x 2 AGQ-S; [2]), despite the 3 x 2 model demonstrating superior data fit [2, 27].”

“Additionally, research has shown that injured athletes returning to sport may be motivated to avoid losing to competitors they used to beat or to avoid losing their spot on the team to a teammate (i.e., other-avoidance); that they may be motivated to surpass their preinjury performance accomplishments or reach performance goals they had set for themselves before the injury (i.e., self-approach); and lastly, that they may be focused on self-based incompetence and thus be motivated to avoid doing worse than they did prior to sustaining their injury (i.e., self-avoidance) [2, 9].” Lines 167=173

"It seemed logical that athletes who felt competent in their post-injury capabilities would have diminished return concerns following a return to competition (e.g., fear of re-injury interfering with performances, struggles to regain technical skills) and might look forward to and/or find enjoyment in achieving personal goals upon return. Therefore…” Line 190-194

“Although previous research suggests that approach-focused achievement goals are associated with experiencing positive processes and outcomes [25], it is possible that such goals might be contraindicated among returning athletes. For example, an intense drive to be the best, to successfully accomplish a task, and/or to improve upon past performances may impel athletes to push themselves too hard toward a premature return to sport, thus increasing the likelihood of sustaining reinjury, a new injury, or decrements in performance. Similarly, although avoidance-focused goals have been linked to maladaptive processes and outcomes [25], it is possible that pursuing avoidance goals may serve as a protective mechanism by motivating athletes to exercise  reasonable caution in order to prevent reinjury and complete a successful return to competition [26].” Lines 198-207.

2. Lines 49 – 50:  This sentence reads, “athletes may experience a variety of positive and negative experiences.” This could be rephrased for better flow.

As suggested, this sentence has been revised on line 50-51 where we state: “Research on the return to competition also reveals that athletes may experience a mixture of adaptive and maladaptive outcomes once they return to competition following injury recovery.”

3.Please clarify the recruitment strategy – were these athletes identified from a single university program (my assumption is they were), and if so, were they treated through the same medical clinic? There could be systematic environmental factors in this setting that would contribute to potential selection biases (eg, self-referral) or measurement biases (eg, supportive therapists promoting high achievement goal or competence scores), leading to over- or under-estimation of the magnitude of the observed relationships. A comment about this in the methods would help the reader assess the likelihood of such issues and give further insight into the generalisability of the results.

Thank you for raising this point. Athletes were recruited from multiple universities, suggesting the generalizability of the findings, a point clarified in the updated manuscript on line 255-256.

4. Lines 315-316: are means of 5.2 and 4.1 really different on a 7-point scale? Given that participants respond using whole numbers and no estimates of variability are presented around these means, I’m not convinced that participants experienced meaningfully higher levels of renewed perspective vs concern. I suggest including a measure of variability here – SD or confidence intervals – and specifying whether this is a statistically vs practically meaningful difference.

As requested, we have provided the SDs for these means and ran a paired t-test that showed statistical differences between these two means. This has now been reported (lines 295-298).

5. Line 321: “positively” should be “positive”

As requested, this change made on line 303.

6. The mediation model accounts for roughly one quarter of the total variance in renewed perspective. In the discussion it would be useful to consider what else might be involved in that relationship and the remaining 75%.

As suggested, we have addressed the issue of remaining variance on lines 433-438.

7. The conclusion makes quite a point about the benefit of using language focused on attaining success vs avoiding failure. This is a valid suggestion but without understanding how social support and intrinsic vs extrinsic motivations interact when establishing perceived competence and/or goal orientations, this is a very specific bit of advice. Language choice is, as the authors have pointed out, one suggestion amongst many empirically-supported intervention techniques and it would be prudent to include the evidence informed ones (goal setting, imagery, etc) in the take home message as well.

As suggested, we have included mention of other empirically-supported interventions as viable options for promoting task-oriented approach goals on lines 532-533.

Reviewer 3 Report

Thank you for the opportunity to review manuscript ID ijerph-782626 entitled ‘Perceived competence, achievement goals, and return-to-sport outcomes: A mediation analysis’ which was submitted for consideration for publication to the International Journal of Environmental Research and Public Health.

This study reports the potential mediating effects of achievement on goals on perceived competence and return to sport outcomes among college athletes after sustaining a sport injury. On the whole the study is well-designed and written. I have some general comments and then some specific comments which I hope will be of assistance to the authors. I wish them all the best in getting their study published.

General comments

Suggest somewhere in abstract and also in aims or methods that the area of focus of the study (i.e. USA) be explicitly stated. Would explain some of the sports and terminology used in this study to an international audience.

Specific comments

Abstract

Line 21 – I find it odd that there are references in the abstract and initially thought these numbers related to 3 components of the research. It wasn’t until I got to the introduction and the references started at 4, that I realised they were references. I will leave it up to the journal editors to determine whether this is allowable within the journal guidelines or not.

Introduction

Extremely detailed, appropriately referenced and clearly identifies a gap in the literature that this study explores.

Methods

Line 225 – explain NCAA and NAIA acronyms for an international audience

Lines 231-233 – suggest a sentence or two explaining these different divisions and also junior college to an international reader. I have no idea what these mean and how they relate to each other.

Line 273 – Is ‘Institutional Review Board’ ethics approval for human subject research? If so state this and add approval number.

Results

No comments

Discussion

Line 483 – this sentence doesn’t make sense – revise

Author Response

Comments and Suggestions for Authors

Thank you for the opportunity to review manuscript ID ijerph-782626 entitled ‘Perceived competence, achievement goals, and return-to-sport outcomes: A mediation analysis’ which was submitted for consideration for publication to the International Journal of Environmental Research and Public Health.

This study reports the potential mediating effects of achievement on goals on perceived competence and return to sport outcomes among college athletes after sustaining a sport injury. On the whole the study is well-designed and written. I have some general comments and then some specific comments which I hope will be of assistance to the authors. I wish them all the best in getting their study published.

Thank you for the positive comments on the manuscript. We address each of the comments below and in the manuscript. Changes appear in red in the revised manuscript.

General comments

Suggest somewhere in abstract and also in aims or methods that the area of focus of the study (i.e. USA) be explicitly stated. Would explain some of the sports and terminology used in this study to an international audience.

As suggested, we indicated that participants were from the United States on lines 20 of the abstract and line 191 of the manuscript. We also explained the terms NCAA and NAIA on line 196-200.

Specific comments

Abstract

Line 21 – I find it odd that there are references in the abstract and initially thought these numbers related to 3 components of the research. It wasn’t until I got to the introduction and the references started at 4, that I realised they were references. I will leave it up to the journal editors to determine whether this is allowable within the journal guidelines or not.

Thank you for the comment. We believe use of references is in the abstract is consistent with the journal guidelines and have left them as is.

Introduction

Extremely detailed, appropriately referenced and clearly identifies a gap in the literature that this study explores.

Thank you.

Methods

Line 225 – explain NCAA and NAIA acronyms for an international audience.

As indicated, we have explained these acronyms on lines 196-200.

Lines 231-233 – suggest a sentence or two explaining these different divisions and also junior college to an international reader. I have no idea what these mean and how they relate to each other.

As requested, we have explained the different divisions on lines 200-204. We also explain the meaning of junior college on lines 211-213.

Line 273 – Is ‘Institutional Review Board’ ethics approval for human subject research? If so state this and add approval number.

Yes, IRB approval is the same as ethics approval. We have indicated this on line 253 and added in the approval number, #00086873.

Results

No comments

Discussion

Line 483 – this sentence doesn’t make sense – revise

As requested, we have revised the language of the sentence on lines 470-471 indicating: “Findings regarding the benefits of task approach goals indicate that coaches, sport medicine practitioners, and health care providers can help injured athletes by adopting language that: (a) orients athletes towards attaining success as opposed to avoiding failure; (b) emphasizes effort, task completion, and correct form; and (c) avoids comparing athletes to others or to their preinjury standards of performance.”